# *Eugenia uniflora* L. Silver and Gold Nanoparticle Synthesis, Characterization, and Evaluation of the Photoreduction Process in Antimicrobial Activities

**DOI:** 10.3390/microorganisms10050999

**Published:** 2022-05-10

**Authors:** Marcia Regina Franzolin, Daniella dos Santos Courrol, Susana de Souza Barreto, Lilia Coronato Courrol

**Affiliations:** 1Laboratório de Bacteriologia, Instituto Butantan, São Paulo 05503-900, Brazil; marcia.franzolin@butantan.gov.br (M.R.F.); daniella.courrol@butantan.gov.br (D.d.S.C.); susana.barreto@butantan.gov.br (S.d.S.B.); 2Departamento de Física, Instituto de Ciências Ambientais, Químicas e Farmacêuticas, Universidade Federal de São Paulo, Diadema 09972-270, Brazil

**Keywords:** *Eugenia uniflora*, pitanga, silver nanoparticles, gold nanoparticles, photoreduction, antimicrobial activity

## Abstract

*Eugenia uniflora linnaeus*, known as Brazilian cherry, is widely distributed in Brazil, Argentina, Uruguay, and Paraguay. *E. uniflora* L. extracts contain phenolic compounds, such as flavonoids, tannins, triterpenes, and sesquiterpenes. The antimicrobial action of essential oils has been attributed to their compositions of bioactive compounds, such as sesquiterpenes. In this paper, the fruit extract of *E. uniflora* was used to synthesize silver and gold nanoparticles. The nanoparticles were characterized by UV–Vis, transmission electron microscopy, elemental analysis, FTIR, and Zeta potential measurement. The silver and gold nanoparticles prepared with fruit extracts presented sizes of ~32 nm and 11 nm (diameter), respectively, and Zeta potentials of −22 mV and −14 mV. The antimicrobial tests were performed with Gram-negative and Gram-positive bacteria and *Candida albicans*. The growth inhibition of *Eu*AgNPs prepared with and without photoreduction showed the important functional groups in the antimicrobial activity.

## 1. Introduction

Antibiotic abuse and overuse have aided the development and spread of resistance mechanisms among bacteria, resulting in multidrug-resistant microorganisms. As a result, alternative therapeutic approaches for microbial pathogens are required.

Plants have biomolecules that are sources of antimicrobial and antifungal compounds that can serve as both reducing and stabilizing agents during the production of silver and gold nanoparticles [1,2,3,4].

*Eugenia uniflora Linnaeus* belongs to the Myrtaceae family, with 142 genera and 5500 species. *E. uniflora* L. is popularly called pitangueira, pitanga, or pitanga-vermelha; the name is derived from the Tupi “pi’tãg”, which means red, in allusion to the color of its fruit [5,6]. The composition of *E. uniflora* L. extracts include phenolic compounds, such as flavonoids, terpenes, tannins, anthraquinones, anthocyanins, flavonoids, carotenoids, and essential oils [7,8,9,10].

The leaves and fruits (pitanga) of *E. uniflora* L. have many therapeutic properties, including antimicrobial, antifungal, antiviral, anti-helminth, insecticide, antidiarrheal, antihypertensive, antitumor, and anti-rheumatic effects [11,12,13,14,15]. Tannins present in *E. uniflora* L. have activity against a broad spectrum of viruses: enteroviruses, caliciviruses, rotavirus, influenza virus A, rhabdo-virus, paramyxoviruses, human immunodeficiency virus, herpes simplex virus, and adenoviruses [16,17,18], and could be used in the fight against COVID-19.

*E. uniflora* were tested against ATCC and clinical isolates of Gram-positive and Gram-negative bacteria and yeast-like fungi [19]. *E. uniflora* essential oil evaluated against strains of several microorganisms showed MIC values ranging from 7 to 100 µg/mL [14,15,20]. The antimicrobial activity observed has been attributed to the presence of different bioactive compounds, such as alkaloids, lignins, phenolic compounds, and terpenoids, that impact the growth and metabolism of microorganisms [7,21].

Chemical synthesis methods lead to the presence of toxic chemical species adsorbed on the surface of nanoparticles. Green methods using plant components, such as roots, rhizomes, stems, leaves, seeds, flowers, fruits, and fruit peel, are inexpensive and eco-friendly [22,23,24,25,26]. Green synthesis of nanoparticles using plant extracts offers advantages over other biogenic methods because it can be suitably applied to large-scale synthesis.

The physicochemical properties of nanoparticles can be easily controlled in the photoreduction process [27,28]. However, a few studies are devoted to using the photoreduction process associated with plant-mediated metallic nanoparticle synthesis [27]. The main advantages of this method are the high spatial resolution, excellent versatility, and local control of reducing agents [29].

In this paper, we have employed the photoreduction method to synthesize silver (*Eu*AgNPs) and gold *(Eu*AuNPs) nanoparticles prepared using the fruit extracts of *Eugenia uniflora* L. The physicochemical and antimicrobial properties of synthesized nanoparticles were investigated.

## 2. Materials and Methods

### 2.1. Materials and Synthesis

*E. uniflora* L. fruits were collected from a spontaneous germination tree in São Paulo, SP, Brazil. The fruits collected in the autumn season were washed with distilled water and chopped into approximately 3 × 3 mm pieces. The chopped material was heated in 100 mL of distilled water until it reached a temperature of 80 °C (±2 °C). The material was filtered, yielding a light-pink liquid.

Immediately after the extract solution filtration, the silver nanoparticle suspensions (*Eu*AgNPs) were prepared by mixing rapidly 40 mL of the extract solution in silver nitrate (AgNO_3_, 1 mmol; Sigma-Aldrich, St. Louis, MO, USA). The color of the solution changed to light brown, indicating the formation of silver nanoparticles. The gold nanoparticle solutions (*Eu*AuNPs) were prepared by mixing 40 mL of the extract solution obtained after filtration with HAuCl_4_ (1 mmol; Sigma-Aldrich). The color of the solution did not present a change, indicating that this step did not produce gold nanoparticles.

Silver and gold nanoparticle solutions (40 mL each) were illuminated with a 300-watt Cermax xenon lamp for 1 min. The Xe lamp was positioned 10 cm from the recipient containing the solution. The illuminated region covered exactly the recipient diameter, and the intensity of the solution was estimated to be 3.6 W/cm^2^. The photoreduction reaction intensified the color of AgNPs and produced a hot-pink color in the AuNPs.

After the reaction, the pH of the solutions was adjusted to ~7.0 using sodium hydroxide. Figure 1 shows some pictures of the synthesis process.

### 2.2. Physicochemical Characterization

A Shimatzu MultiSpec 1501 spectrophotometer was used to measure the UV–Vis region absorption spectra. For the measurement, 50 µL of NPs were diluted in 500 µL of double-distilled water. The measurements were performed in a 10 mm optical path quartz cuvette in the range of 200 and 800 nm.

The FTIR (Fourier-transform infrared spectroscopy) was obtained with a Shimatzu IRPrestige. The aliquots of 200 µL of the extracts and the NPs were dried at 60 °C for approximately 20 min. The procedure was repeated three times to obtain sufficient material to prepare KBr pellets. The temperature influence on the stability of nanoparticles was checked by submitting 10 mL of *Eu*Ag and *Eu*AuNPs to temperatures of 60 °C for one hour.

The colloidal suspensions were analyzed by Zeta potential measurement using the Zetasizer Nano ZS Malvern apparatus. Three sizes were made for each sample.

Transmission electron microscopy (TEM) images of prepared nanoparticles were obtained. The samples were dripped onto a copper grid and analyzed on a JEM 2100—JEOL transmission electron microscope. Particle size distribution analysis was performed using images obtained by this technique with ImageJ software.

Energy dispersive spectroscopy (EDS) was performed with a JSM-7610F JEOL scanning electron microscope (SEM) which obtained electron microscopy images. The samples (100 mL) were deposited in an aluminum support and dried in an oven at 60 °C for 30 min.

### 2.3. Microorganism Growth Inhibition

The *Eu*AgNPs and *Eu*AuNPs, were evaluated against Gram-negative bacteria, namely, *Escherichia coli* American Type Culture Collection (ATCC) 25922, *Escherichia coli* O44:H18 EAEC042 [30] (clinical isolate), *Klebsiella pneumoniae* ATCC 700603, *Pseudomonas aeruginosa* ATCC 27853, *Salmonella*
*Typhimurium* ATCC 14028, Gram-positive bacteria *Bacillus subtilis* ATCC 6633, clinical isolates of methicillin-resistant *Staphylococcus aureus* (MRSA) and *Enterococcus faecalis*, *Staphylococcus aureus* ATCC 25923, and the yeast *Candida albicans* ATCC 10231. The clinical strains present the following resistance profile: *E. coli* O44:H18 EAEC042—chloramphenicol and tetracycline; MRSA—amikacin, streptomycin, clindamycin, gentamicin, oxacillin, penicillin, and sulfonamide; and *E. faecalis*—amikacin, clindamycin, gentamicin, oxacillin, penicillin, and erythromycin.

The antimicrobial activities of *Eu*NPs prepared without photoreduction (*Eu*AgNPs) and synthesized with photoreduction (*Eu*AgNPs (PR) and *Eu*AuNPs (PR)) were investigated according to the CLSI guidelines. Bacterial (Mueller–Hinton (MH) broth) and fungal inoculum (Sabouraud (Sab) broth, *C. albicans*) were adjusted to approximately 10^6^ CFU/mL. Samples were incubated with 50µL of culture medium or 50 µL of NP solutions diluted ten times in MH or Sab broth, resulting in a final volume of 100 µL, with 10^4^ CFU/well and a final dilution of solutions equal to 20 times. Antimicrobial analysis of the extracts was performed with a dilution of 20 times to evaluate the antimicrobial activity of the nanoparticles. After incubation at 37 °C for 20 h, the microbial growth optical density (OD) at 595 nm was measured in an enzyme-linked immunosorbent assay (ELISA) reader (Multiskan^®^EX; Thermo Fisher Scientific, Waltham, MA, USA). The growth inhibition (%) results were calculated using the following equation:$$\%\mathrm{Cell}\mathrm{inhibition}=\frac{\mathrm{Control}\mathrm{at}\;OD_{595\mathrm{nm}}-\mathrm{Test}\mathrm{at}\;OD_{595\mathrm{nm}}}{\mathrm{Control}\mathrm{at}\;OD_{595\mathrm{nm}}}\times 100$$
where $\mathrm{Control}\mathrm{at}\;OD_{595\mathrm{nm}}$ is the absorbance value (OD) of control (microorganisms in the absence of nanoparticles) and $\mathrm{Test}\mathrm{at}\;OD_{595\mathrm{nm}}\;$is the absorbance value (OD) of microorganisms incubated with nanoparticles. %Cellinhibition is expressed as means ± standard deviation of triplicate assays.

### 2.4. Statistical Analysis

GraphPad Prism 4.00 software was used for statistical analysis. Results with *p* < 0.05 according to the Student’s *t*-test were considered significant.

## 3. Results

### 3.1. Synthesis and Characterizations

Figure 2a shows the UV–Vis spectra of fruit extracts and multi-curve fitting (green line). Bands around 248, 304, 371 nm, probably due to quercetin [31], and around 462 nm, due to carotenoids [32], can be observed.

Figure 2b shows the UV–Vis spectrum of *Eu*AgNPs. For silver nanoparticles prepared without photoreduction, a peak around 444 nm was observed due to surface plasmon resonance (SPR), similar to the results obtained by Dugganaboyana et al. [33]. For *Eu*AgNPs prepared by the photoreduction process, the SPR band shifted to ~422 nm and became narrow, indicating that monodispersed nanoparticles were produced.

For *Eu*AuNPs synthesized by the photoreduction process, the gold nanoparticles’ SPR band was observed around 535 nm [34], as shown in Figure 2c.

The *Eu*AgNPs prepared by the photoreduction process were spherical, with sizes ~32 nm (Figure 2d), and the *Eu*AuNPs had spherical shapes and smaller sizes of ~11 nm (Figure 2e). The EDS study for *Eu*AgNPs and *Eu*AuNps estimated that the NPs present Ag and Au, respectively, and bonds including C, O, Cu, Mn, and Mg were also identified, corresponding to a small percentage of the total mass.

The respective values for the Zeta potential measurement and the polydispersity index (PI) were −21.7 mV and 0.369 for *Eu*AgNPs and −13.7 mV and 0.451 for *Eu*AuNPs.

Figure 3 presents the Fourier-transform infrared spectroscopy (FTIR) obtained for *Eu*NPs. Bands around 3400 cm^−1^ related to -NH and bonded -OH groups of carboxylic acids [35] can be observed. The bands around 2980–2800 cm^−1^ indicate the presence of C-H_2_ asymmetric and symmetric stretching. The bands in the 1600 and 1700 cm^−1^ region are due to the C=C stretching vibration of aromatic rings [4,22], the vibration of N-H of amines, C=O of amides, and carboxylic groups [25]; in addition, the band around 1633 cm^−1^ is related to flavonoids and amino acids: ν(C=O), ν(C=C), and δas(N-H) [23].

A significant difference between silver (Figure 3b) and gold nanoparticles (Figure 3a) was observed. The peak observed around 1384 cm^−1^, correspondent to the C-H bending of aldehyde groups from the glucose structure, must be responsible for the bioreduction of Ag^+^ and capping/stabilization of silver nanoparticles.

Differences for AgNPs prepared with or without the photoreduction process can be observed in Figure 3b, mainly in the ratios between bands around 1730 and 1600 cm^−1^. The absorption peak at 1724 cm^−1^ observed for NPs submitted to photoreduction is more intense than for the nanoparticles without photoreduction. The peak at 1724 cm^−1^ corresponds to C=O stretch (carbonyl). The presence of this band in the case of plant-mediated AgNPs demonstrates the involvement of phenolic compounds.

Figure 3c verified the influence of temperature on the stability of nanoparticles. The UV–Vis spectra and *Eu*NP solution color remained unchanged after 1 h at 60 °C.

### 3.2. Antimicrobial Tests

The antimicrobial activities of *Eu*AgNPs and *Eu*AuNPs were tested using the broth microdilution assay, which provides quantitative data on inhibition efficacy, and the results obtained and compared with the antimicrobial activities of fruit extracts are presented in Figure 4. The results indicated high percentual inhibition of Gram-positive and Gram-negative bacteria and yeast treated with *Eu*AgNps (54% to 100%, with a mean of 95.12 ± 2.03%), in comparison with *Eu*AuNPs (*p* < 0.05). The antimicrobial effect of *Eu*AuNPs ranged from 0–71.27%, presenting a mean of 32.39 ± 5.02. *Eu*AgNPs without photoreduction and with photoreduction did not show a significant (*p* > 0.05) difference. Most of the microbial species, including *C. albicans*, showed high percentual inhibition (>90%) when exposed to *Eu*AgNPs prepared with photoreduction, except for *E. faecalis* (53.71%).

All treatments tested exhibited high percentual inhibition of *B. subtilis*, especially *Eu*AgNPs (~99%), whereas the percentual inhibition for *Eu*AuNPs reached ~71%. On the other hand, *Eu*AuNPs were ineffective against *K. pneumoniae* ATCC 700603.

The inhibition of fruit extracts was lower than 20%.

## 4. Discussion

Phenolic substances present in plant extracts have in their structure one (or more) aromatic rings with one or more hydroxyl substitutes [36]. Phenolic metabolites include phenolic acids, flavonoids, simple phenolics, phenylpropanoids, coumarins, tannins, and tocopherols [5].

In general, phenolic substances are potent antioxidants which act by several mechanisms, such as electron donation and interruption of chains of oxidation reactions [36]. In addition to their antioxidant capacities, phenolic substances—mainly phenolic acids, flavonoids, and tannins—have health-beneficial properties, such as anticancer, antimicrobial, antiallergic, hepatoprotective, antithrombotic, antiviral, vasodilator, antimutagenic and anti-inflammatory activities, many of these biological functions having been correlated with their antioxidant capacities.

The absorption spectra for the aqueous extracts of *E. uniflora* L. fruits (Figure 2a) show absorbance peaks in the ultraviolet region at wavelengths compatible with phenolic substances [35]. Absorption bands around 248, 304, and 371 nm can be attributed to flavonoids, such as quercetin [37], and the peak around 462 nm is probably due to carotenoids.

Exploring plant extracts for nanoparticle synthesis makes the procedure a green method and economically viable [38,39,40,41]. Several works in the literature evidence good results for the synthesis and antimicrobial activities of nanoparticles prepared with plant extracts [38,39,40,41]. Table 1 shows some results obtained by other authors using fruit extracts for antimicrobial purposes. This paper discusses *Eu*NPs with fruit extracts associated with the photoreduction method and pH control.

*E. uniflora* L. phytochemicals, such as saponin, tannins, phenolic compounds, and flavonoids [51,52], interact with AgNO_3_ reduced to elemental silver or gold. Biosynthesis probably involves quercetin [53], carotenoids, and luteolin [54]. The presence of silver nanoparticles is confirmed in Figure 2b. Immediately after mixing plant extracts and silver nitrate, the phytochemical agents reduce silver and inhibit the ion agglomerations to produce nanoparticles. The SPR bands are wide, indicating the presence of agglomerates. Although electrostatic stabilization is easier to maintain in colloidal media, it is impossible to separate agglomerated particles due to strong forces of interactions between oppositely charged ions [55]. The photoreduction process and pH adjustment improve the optical properties of *E. uniflora* L. silver nanoparticles. Photoreduction offers steric repulsion within nanoparticles, thus preventing agglomeration and giving rise to a mutual stabilization system [56]. The production of gold nanoparticles with fruit extract was possible due to the photoreduction process that reduced gold ions to Au^0^. Without this effect, no change in solution color was observed after mixing the extract with HAuCl_4_.

Silver nanoparticles presented a size of around 32 nm, whereas the size of gold nanoparticles ranged around 11 nm. The Zeta potential results indicated negatively charged nanoparticles.

It is known that phenolic substances (flavonoids and non-flavonoids), among others, are responsible for the antimicrobial properties of plants. According to Sobeh et al. [57], the essential oil extracted from *E. uniflora* L. presents antimicrobial activity against *S. aureus*, *S. epidermidis*, *Bacillus licheniformis*, *B. subtilis*, *E. faecalis*, *E. coli*, *K. pneumoniae*, *P. aeruginosa*, *Candida parapsilosis*, and *C. albicans*.

The results obtained for *Eu*AgNPs, shown in Figure 4, indicated that low concentrations (final dilutions of NP solutions equal to 20 times) exhibited high inhibitory activity against Gram-positive and Gram-negative bacteria. Fruit extracts presented a maximum inhibition of ~18% with the tested dilution. In the case of nanoparticles prepared with plant extracts, the phytochemicals present on their surface can be delivered quickly inside microorganisms, issuing in the cytotoxic effects of the proper silver and gold ions.

All tested microorganisms treated with *Eu*AgNPs prepared with and without photoreduction showed inhibition of >90% without significant differences, except for *E. faecalis*, which presented a 53.61% of inhibition with photoreduction. The advantage of the photoreduction process is the improvement of physical characteristics and the stability of the nanoparticles. *Eu*NPs prepared by photoreduction keep their color for at least one year.

The outer membranes of Gram-negative bacteria contain proteins, phospholipids, and lipopolysaccharides that serve as a barrier to the external environment and enable selective diffusion through porins, allowing uptake and waste removal. Gram-negative bacteria present increased adhesion on positively charged layers compared to negatively charged layers [55].

Our results obtained with FTIR (Figure 3b) showed that the main differences in nanoparticles prepared without and with photoreduction were observed in the ratios of lines in the 1600 cm^−1^ region, with the stretching of carboxyl groups, and, in the 1724 cm^−1^ region, with the stretching of the C=O bonds of ester carbonyl groups, probably attributable to the presence of quercetin [56]. Nanoparticles synthesized without photoreduction have more carbonyl groups in their structures. The carbonyl bond is highly polar, with a partial positive charge on the carbon and a partial negative charge on the oxygen. When the carbonyl bond of a nanoparticle structure meets a specific anionic bacterial surface (carboxylic residues, phosphate residues, etc.), electrostatic binding can facilitate uptake, which could explain the better results obtained for *E. faecalis* and MRSA for the nanoparticles synthesized without the photoreduction process. The nanoparticles prepared by the photoreduction process have relatively more carboxylic groups with negative charges and could have fewer interactions with Gram-negative microorganisms.

*Eu*AuNPs presented low antimicrobial inhibition, but *B. subtilis* treated with AuNPs demonstrated inhibition >70%. The results showed that *Eu*AgNPs exhibited superior antimicrobial activity relative to *Eu*AuNPs when tested against microbial strains, consistent with the results of other authors who observed only bacteriostatic activities with gold nanoparticles [58,59].

Metal nanoparticles can penetrate bacterial cells and interact with sulfur and phosphorous bases from DNA molecules, decreasing the capacity for cell replication. Another mechanism of inhibitory action is the induction of oxidative stress due to the generation of reactive oxygen species (ROS), including free radicals. These can damage the cell membrane, make it porous, denature proteins, and inhibit cellular respiratory enzymes, leading to cell death [60,61,62]. *Eu*NPs may employ one or more of these mechanisms to inhibit microbial growth.

The facile synthesis and excellent properties, such as good stability and low polydispersity, guarantee several essential applications in biology and medicine for the synthesized nanoparticles, which showed a broad spectrum of antimicrobial activity. These nanoparticles could be impregnated in equipment for individual protection and reduce bacterial growth on devices, reducing the number of hospital-acquired infections.

## 5. Conclusions

*Eu*NPs were successfully prepared using aqueous fruit extract as a reducing agent. The synthesized *Eu*NPs were characterized thoroughly by UV–Visible, FTIR, Zeta potential measurement, TEM, and SEM. The *Eu*NPs were monodispersed, stable, and spherical, with sizes of ~32 nm for AgNPs and 11 nm for AuNPs. The FTIR results found differences between nanoparticles prepared with and without a photoreduction process. Nanoparticles synthesized without the photoreduction process presented more carbonyl groups in their structures than those prepared with a photoreduction process, which showed relatively more carboxylic groups. Furthermore, this study demonstrated that bacterial growths of Gram-positive and -negative bacteria and *C. albicans* were inhibited by the synthesized AgNPs, while low inhibition was observed for the gold nanoparticles. The silver nanoparticles prepared with photoreduction showed elevated antimicrobial activity against Gram-positive and -negative bacteria and *C. albicans* and better physical properties than nanoparticles prepared without photoreduction process.

## Figures and Tables

**Figure 1 microorganisms-10-00999-f001:**
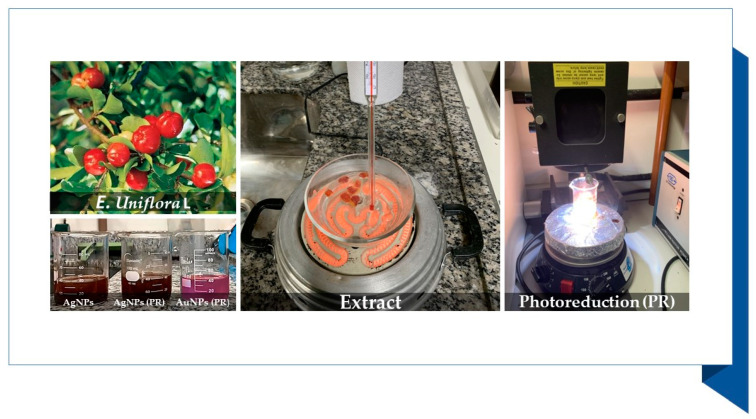
Nanoparticles prepared with *E. uniflora* extracts: obtainment of the extract and the photoreduction process.

**Figure 2 microorganisms-10-00999-f002:**
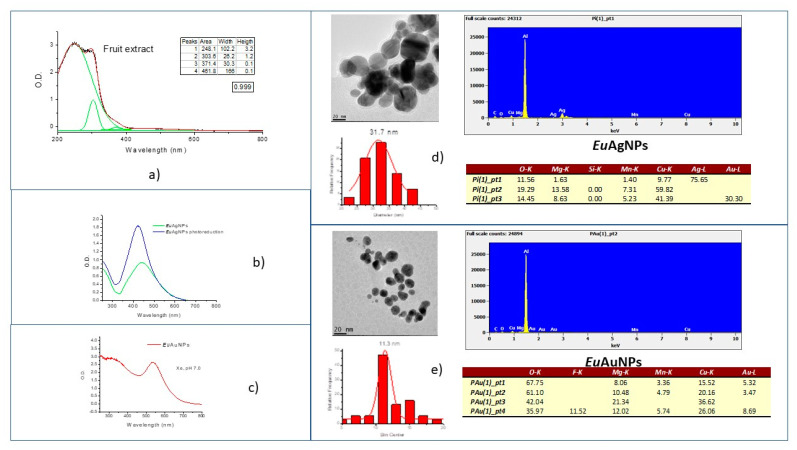
UV–Vis absorbance spectra of: (**a**) *E. uniflora* L. fruits extracts (green lines represent the multi-curve fitting in which peaks are described in the tables); (**b**) *Eu*AgNPs prepared with and without photoreduction; (**c**) *Eu*AuNPs prepared with fruits extracts by photoreduction. (**d**) TEM image and elemental analysis for *Eu*AgNPs and (**e**) for *Eu*AuNPs (photoreduction).

**Figure 3 microorganisms-10-00999-f003:**
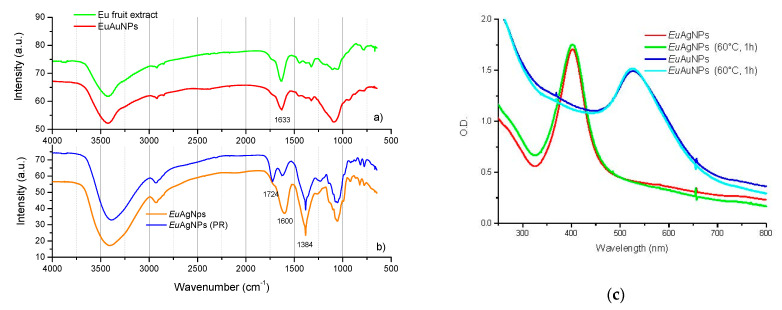
FTIR spectra of: (**a**) *Eu* fruit extract and *Eu*AuNPs; and (**b**) *Eu*AgNPs prepared without and with photoreduction (PR). (**c**) Influence of temperature on the stability of the nanoparticles.

**Figure 4 microorganisms-10-00999-f004:**
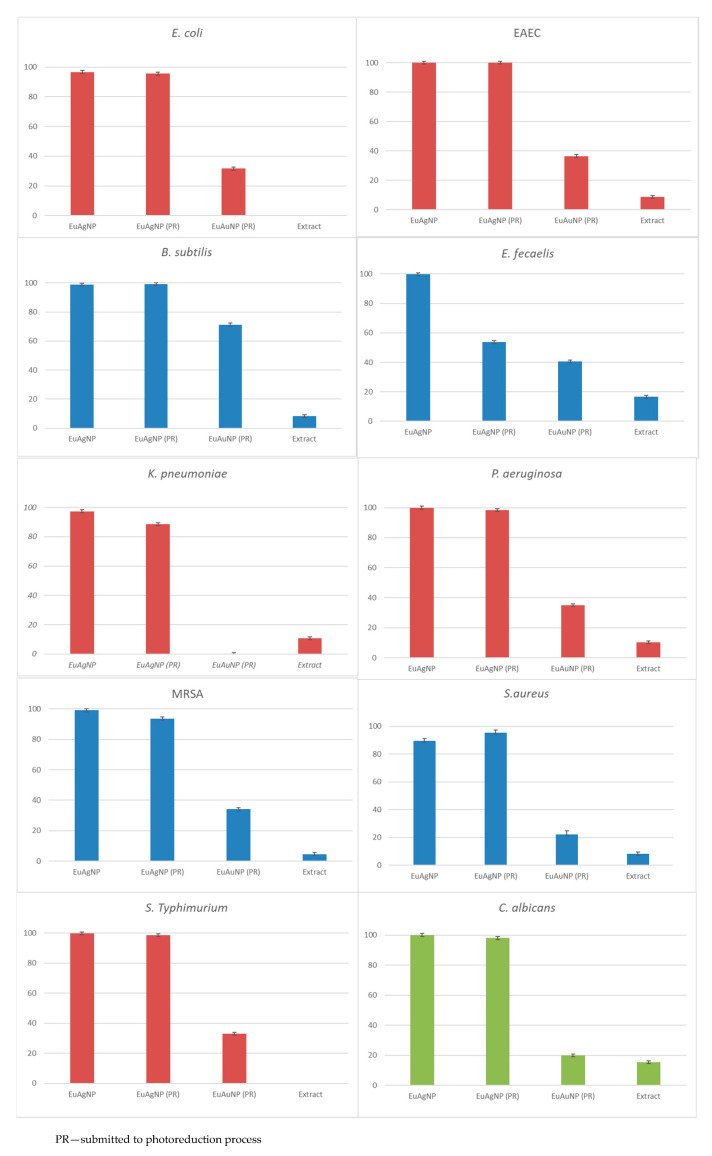
Antimicrobial growth inhibition of *Eu*AgNP, *Eu*AgNP (PR), *Eu*AuNP (PR), and fruit extract against *Escherichia coli* ATCC 25922, *Escherichia coli* O44:H18 EAEC042, *Salmonella*
*Typhimurium* ATCC 14028, *Klebsiella pneumoniae* ATCC 700603, *Pseudomonas aeruginosa* ATCC 27853, *Bacillus subtilis* ATCC 6633, *Staphylococcus aureus* ATCC 25923, methicillin-resistant *Staphylococcus aureus* (MRSA), *Enterococcus faecalis*, and *Candida albicans* ATCC 10231. PR: Photoreduction.

**Table 1 microorganisms-10-00999-t001:** Fruit-derived NPs with antimicrobial activities.

Fruits Extract	Medium	Nanoparticles Properties	Microorganisms Tested	Ref.
*Nauclea latifolia* (African peach)	Methanol and water	AgNPs: 600 nm	*C freundii*, *EC 35218*, *E. coli 11775*, *S. aureus 29213*, *E. coli, C. albicans*, *Rhizopus*, *Klebsiella* sp., and *Staphylococcus* sp.	[42]
*P. americana* (Butter fruit)	Methanol	AgNPs: 420 nm Size: 20–50 nm	*P. vermicola*	[43]
*Artocarpus lakoocha*	Methanol	AgNPs: 300–700 nm Size: 6.59–25 nm	*S. pneumoniae*, *S. aureus*, *K. pneumoniae*, *B. subtilis*, *E. coli*, and *S. flexneri*	[44]
*Sorbus aucuparia* (Rowanberries)	Water	AgNPs: 300–700 nm Size: 20–30 nm Zeta: −28.8 mV AuNPs: 500–600 nm Zeta: −25.6 mV 90–100 nm	*E. coli UTI 89* and *P. aeruginosa PAO1.*	[45]
*Citrus limon*	Juice	AuNPs: 575 nm Size: 30 ± 6 nm	*K. pneumoniae* and *Listeria monocytogenes*	[46]
Ripe fruit of *Crescentia alata Kunth*, *Vitex mollis Kunth*, and *Randia echinocarpa Sessé et Mociño*	Water	AgNPs: *V. mollis* (435 nm), *C. alata* (416 nm), and *R. echinocarpa* (412 nm) Size: 13–31 nm AuNPs: *V. mollis* (510 nm) Size: 2–16 nm	*Streptococcus group A-4*, *S. aureus 3*, *E. coli A011*, *E. coli A055, S.**aureus ATCC 29213*, *Shigella dysenteriae*, *P. aeruginosa ATCC 27853*, and *E. coli ATCC 25922*	[47]
*A. villosum*	Water	AgNPs: 428 nm Size: 5–15 nm, PDI: 0.246 AuNPs: 550 nm Size: 5–10 nm, PDI: 0.237	*S. aureus* and *E. coli*	[48]
*Phoenix dactylifera* (Palm tree)	Water	AgNPs: 395–425 nm, Size: 25–60 nm, Zeta: −35 mV	*Bacillus cereus*, *S. aureus*, *Staphylococcus epidermidis*, *K. pneumoniae*, and *E. coli*	[49]
*Banana peel*	Water	AgNPs: 430 nm	*C. albicans BH*, *Shigella* sp., *Klebsiella* sp., and *Citrobacter kosari*	[50]
*E. uniflora* (Pitanga)	Water	AgNPs: 422 nm Size: ~32 nm Zeta: −22 mV PDI: 0.369 AuNPs: 535 nm Size: ~11 nm Zeta: −14 mV PDI: 0.451	*E. coli ATCC 25922, E. coli* O44:H18 EAEC042 (clinical isolate), *B.* *subtilis* ATCC 6633, *K. pneumoniae* ATCC 700603, *Pe aeruginosa* ATCC 27853, *Se Typhimurium* ATCC 14028, *S. aureus* ATCC 25923 and clinical isolates of methicillin-resistant *Se aureus* (MRSA), and *E. faecalis*, and *C. albicans* ATCC 10231	This work

## Data Availability

Not applicable.

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
