# Peer review of "Eugenia uniflora L. Silver and Gold Nanoparticle Synthesis, Characterization, and Evaluation of the Photoreduction Process in Antimicrobial Activities"

_microorganisms, 2022, doi:10.3390/microorganisms10050999_

Round 1

Reviewer 1 Report

The manuscript entitled "Eugenia uniflora L. Silver and Gold Nanoparticles Synthesis, Characterization, and Evaluation of Photoreduction Process in Antimicrobial Activities" presents the results of a study on the "green" synthesis and research of the characteristics of silver and gold nanoparticles. This study was carried out on an actual topic, many similar studies have been published in which extracts of other plant species are used. The article is written concisely, competently and consistently, and can be accepted after minor changes.
1) There is a lot of free space on page 4, which can be filled with text.
2) Figure 4. The signature uses a different font.
3) The authors could add in the review of literary sources a reference to https://doi.org/10.3390/mi12121480.

Author Response

The authors would like to thank the reviewer's questions and comments.

1)    There is a lot of free space on page 4, which can be filled with text.

Spaces were filled.

2)    Figure 4. The signature uses a different font.

The font was corrected.

3)    The authors could add in the review of literary sources a reference to https://doi.org/10.3390/mi12121480.

The reference was included.

Author Response

The Referee’s comments were valuable and gave us an excellent opportunity to improve the text.

1. In section 2.1. Materials and synthesis- the description of the experimental procedure of the synthesis of the nanoparticles should say how the addition of the silver and gold salt was done: was it added in solution? Rapidly, all at once? Slowly?

Section 2.1 was changed to clarify the synthesis procedure.

2. In section 2.2. Physicochemical Characterization- it is stated that the colloidal stability is evaluated by determination of the zeta potential of the nanoparticles. Several questions arise from this procedure:

(a) The zeta potential is not sufficient by itself to assess the colloidal stability of nanoparticles. it should be conjugated with other measurements namely by UV-vis and by PXRD.

UV-Vis spectra and the color of nanoparticles synthesized by photoreduction method remained unchanged for at least one year after the synthesis by the photoreduction method, indicating high stability.

 (b) Considering that the stability of nanoparticles is only determined by the zeta potential, it must be demonstrated that the stability is maintained over time and concretely over the period of time necessary to evaluate the antimicrobial activity.

The antimicrobial tests were performed one week after the synthesis of nanoparticles. During the experiment, the color and the UV-Vis spectra of nanoparticles was unchanged, and no reaction with the culture medium was observed.

 (c) in addition, a study of the effect of temperature on the stability of the nanoparticles should be done. The infrared spectrum was obtained after the nanoparticles were subjected to temperatures of 60°C. The authors draw conclusions based on this study, without having determined the stability of the nanoparticles after this procedure.

The authors thank the revisor for this interesting question. The influence of temperature on the stability of nanoparticles was verified by submiting 10 mL of EuAg and EuAuNPs at 60°C for one h. The UV-Vis spectra and solution color remained the same after this period proving the stability. This result was included in the manuscript (Figure 4c).

 3. In section 3.1. Synthesis and Characterization- should be shown in Fig 2 the b) and c) in the corresponding figure.

Figure 2 was changed.

 4.The infrared spectrum corresponding to the uniflora of fruit extract should be indicated.

The infrared spectrum of the fruit extract was included in Figure 4a.

 5. In section 3.2 Antimicrobial Tests it is mandatory that a pooled analysis of the extracts be performed. Only then can the antimicrobial activity of the nanoparticles be evaluated. Could it be that all the activity verified comes from the extracts alone and not from the nanoparticles? Are the nanoparticles stable during the tests? Have stability tests been done in the culture medium?

The antimicrobial activities of the fruit extract in the same dilution of nanoparticles was included in the Figure 5.

The growth inhibition of fruit extracts was lower than 20%.

The nanoparticles were stable during the tests. No color change of nanoparticles was observed in the culture medium, indicating stability.

6. Authors should compare their results with those already described in the literature. There are numerous examples in the literature of evaluations of antimicrobial activity of nanoparticles. The added value of these nanoparticles and in particular the effect of photoreduction on their biological activity should be discussed.

Table 1 showing fruit-derived NPs with antimicrobial activities was added to the manuscript in “Discussion Section”. Our work is the only one reporting nanoparticles prepared by the photoreduction method.

Round 2

Reviewer 2 Report

In my opinion, the authors have taken into account the comments made to them and improved the manuscript. I think that the manuscript can be published in Microorganisms, in this form